# Differences in GIP Receptor Expression by Feeding Status in the Mouse Brain

**DOI:** 10.3390/ijms26031142

**Published:** 2025-01-28

**Authors:** Do Kyeong Song, Narae Jung, Yeon-Ah Sung, Young Sun Hong, Hyejin Lee

**Affiliations:** Department of Internal Medicine, Ewha Womans University School of Medicine, 25, Magokdong-ro 2-gil, Gangseo-gu, Seoul 07804, Republic of Korea; dksong@ewha.ac.kr (D.K.S.); jnr583@naver.com (N.J.); yasung@ewha.ac.kr (Y.-A.S.); imhys@ewha.ac.kr (Y.S.H.)

**Keywords:** GIP (gastric inhibitory polypeptide), hypothalamus, obesity

## Abstract

Gastric inhibitory polypeptide (GIP) contributes to energy metabolism regulation. We investigated differences in GIP receptor expression in the brain by feeding status among lean and obese mice and the effect of acute central GIP administration on the expression of appetite-regulating hypothalamic neuropeptides. We divided the mice into four groups: fed/lean, fasted/lean, fed/obese, and fasted/obese. The arcuate nucleus (ARC), paraventricular nucleus of the hypothalamus, and nucleus of the solitary tract in the brainstem were harvested. GIP (6 nmol) or saline was injected for the acute intracerebroventricular administration test, followed by the collection of hypothalamic tissue after 2 h. Fed/obese mice had higher ARC GIP receptor mRNA levels than fasted/obese and lean mice. This difference was not observed among lean mice by feeding status. Obese mice had higher blood GIP levels than lean mice. Fed/obese mice had higher blood GIP levels than fasted/obese mice. This difference was not observed among lean mice by feeding status. GIP administration significantly increased proopiomelano-cortin (*Pomc*) mRNA levels (GIP: 7.59 ± 0.14; saline: 3.44 ± 1.38 arbitrary units; *p* = 0.030). Increased GIP receptor expression in the ARC in obese mice indicates its central nervous system involvement in energy balance regulation. GIP potentially regulates POMC-mediated appetite regulation in the hypothalamus. It is possible that POMC neurons are targets of GIP action in the brain.

## 1. Introduction

Gastric inhibitory polypeptide (GIP) is a 42-amino acid polypeptide produced by enteroendocrine K cells in the proximal small intestine and secreted into circulation in response to nutrient ingestion. Although GIP is primarily known to mediate the postprandial insulin response, it is considered as the obesity hormone given its role in the regulation of energy metabolism regulation [1,2]. GIP levels are reportedly high in obese patients [3] and in mice with high fat diet (HFD)-induced obesity [4]. The GIP-GIP receptor (GIPR) system is known to significantly contribute to the development of obesity. GIP or GIPR inhibition protects mice from diet-induced obesity, and administration of long-lasting GIP derivatives or transgenic overexpression of GIP has an anti-obesity effect [5,6,7]. GIPR signaling in adipose tissue has been suggested to play an important role in HFD-induced insulin resistance and hepatic steatosis through adipose tissue-specific GIPR knockout mice [8]. In addition, tirzepatide, a glucagon-like peptide-1 (GLP-1)/GIP hybrid peptide, reportedly induces weight loss and is used in clinical settings [9,10]. GIP can paradoxically affect body weight because the inhibition of endogenous GIP action or exogenous administration of supraphysiological doses of GIP has anti-obesity effects.

Although the exact mechanism underlying the anti-obesity effect of GIP or the GIPR system is unclear, cumulative results support that the therapeutic effect associated with the modulation of the GIP system is mediated by the central nervous system (CNS) [1]. Acute intracerebroventricular administration of GIP at supraphysiologic doses yields significant weight reduction in mice [11,12]. Long-lasting GIP agonists reportedly reduce body weight in mice with HFD-induced obesity. The blunted weight-reducing effect of intracerebroventricular administered acyl-GIP in the brain-specific GIPR knockout mice indicates that the anti-obesity effect of GIP or the GIPR system is mediated via GIPR in the CNS. Furthermore, CNS-specific GIPR knockout mice showed lower body weights despite a HFD [12]. Central administration of GIPR-neutralizing antibody showed anti-obesity effects in diet-induced obese mice [13].

GIP receptor mRNA was reported to be present in peripheral organs and the brain including in the pancreas, the gut, the adipose tissue, the heart, the pituitary, and the adrenal cortex [14]. Previous studies suggest that the GIPR is present throughout the brain including within the hypothalamus [15]; however, studies investigating the differences in GIPR expression by feeding status in the mouse brain are lacking. To determine the mechanism by which GIP regulates energy metabolism throughout the body, understanding the regulation of GIP-GIPR signaling in the brain is crucial. We aimed to investigate the differences in GIPR expression in the brain by feeding status in lean and obese mice and to investigate the effect of acute central GIP administration on the expression of hypothalamic neuropeptides involved in appetite regulation in mice.

## 2. Results

### 2.1. Changes in Gipr mRNA Expression in Hypothalamus and Adipose Tissue by Feeding Status

Fed/obese mice had higher arcuate nucleus (ARC) *Gipr* mRNA levels than fasted/obese mice (2.75 ± 0.34 vs. 1.24 ± 0.24 arbitrary units; *p* = 0.009). ARC *Gipr* mRNA levels in fed/obese mice were higher than in fed/lean (2.75 ± 0.34 vs. 1.24 ± 0.24 arbitrary units; *p* = 0.009) and fasted/lean mice (2.75 ± 0.34 vs. 0.84 ± 0.28 arbitrary units; *p* = 0.002). However, ARC *Gipr* mRNA levels did not differ by feeding status in lean mice (Figure 1). For changes in ARC *Gipr* mRNA expression in the hypothalamus, the interaction effect of the two factors (factor 1: obese vs. lean; factor 2: fed vs. fasted) was not significant (*p* = 0.059). The interaction effect between the obesity status (obese vs. lean; *p* = 0.003) and that between the feeding status (fed vs. fasted; *p* = 0.002) appear to independently affect the ARC *Gipr* mRNA levels.

The paraventricular nucleus of the hypothalamus (PVH) *Gipr* mRNA levels did not differ by feeding status both in lean and obese mice (fed/obese: 1.12 ± 0.20; fasted/obese: 0.87 ± 0.18; fed/lean: 1.21 ± 0.17; fasted/lean: 1.54 ± 0.18 arbitrary units; all *p* > 0.05). For changes in PVH *Gipr* mRNA expression in the hypothalamus, the interaction effect of the two factors (factor 1: obese vs. lean; factor 2: fed vs. fasted) was not significant (*p* = 0.226). The PVH *Gipr* mRNA levels showed that the effects between the obese mice and the lean mice (*p* = 0.061) and the effect between the fed mice and the fasted mice (*p* = 0.981) were also not significant.

*Gipr* mRNA levels in the nucleus of the solitary tract (NTS) levels did not differ by feeding status both in lean and obese mice (fed/obese: 0.64 ± 0.30; fasted/obese: 0.69 ± 0.30; fed/lean: 1.80 ± 0.29; fasted/lean: 1.44 ± 0.36 arbitrary units; all *p* > 0.05; Figure 1). For changes in NTS *Gipr* mRNA expression in the hypothalamus, the interaction effect of the two factors (factor 1: obese vs. lean; factor 2: fed vs. fasted) was not significant (*p* = 0.992). For the NTS *Gipr* mRNA expression in the hypothalamus, the effect between the obese mice and the lean mice was found to have a significant effect (*p* = 0.009); however, the effect between the fed mice and the fasted mice was not significant (*p* = 0.754).

Figure 2 shows the expression of *Gipr* mRNA in the mouse adipose tissue by feeding status. Gonadal fat (fed/obese: 4.22 ± 0.55; fed/lean: 1.03 ± 0.63 arbitrary units; *p* = 0.009), brown adipose tissue (fed/obese: 5.19 ± 0.61; fed/lean: 1.31 ± 0.64 arbitrary units; *p* = 0.001), and inguinal fat (fed/obese: 11.18 ± 2.11; fed/lean: 1.81 ± 3.17 arbitrary units; *p* = 0.004) *Gipr* mRNA levels were higher in fed/obese mice than in fed/lean mice. However, adipose tissue *Gipr* mRNA levels did not differ by feeding status both in lean and obese mice (gonadal fat fasted/obese: 3.11 ± 0.37; gonadal fat fasted/lean: 1.15 ± 0.55; brown adipose tissue fasted/obese: 3.06 ± 0.57; brown adipose tissue fasted/lean: 2.63 ± 0.57; inguinal fat fasted/obese: 7.55 ± 2.11; inguinal fat fasted/lean: 2.53 ± 3.66 arbitrary units).

### 2.2. Changes in Plasma GIP Levels by Feeding Status

Fed/obese mice had higher blood GIP levels than fasted/obese mice (954.2 ± 69.6 vs. 669.2 ± 73.4 pg/mL; *p* = 0.049). However, plasma GIP levels did not differ by feeding status in lean mice (fed/lean: 438.9 ± 73.4; fasted/lean: 348.4 ± 73.4 pg/mL; *p* = 1.00; Figure 3).

### 2.3. Changes in Appetite-Regulating Hypothalamic Neuropeptide mRNA Levels After Intracerebroventricular Administration of GIP

Food intake (GIP-treated mice: 0.75 ± 0.31; saline-treated mice: 3.02 ± 0.17 g; *p* < 0.001) and body weight (GIP-treated mice: −3.18 ± 0.33; saline-treated mice: 0.06 ± 0.21 g; *p* < 0.001) more significantly decreased for 24 h in mice after intracerebroventricular administration of GIP than after intracerebroventricular administration of saline (all *p* < 0.05; Figure 4). Figure 5 shows the changes in leptin receptor (*Lepr*), proopiomelanocortin (*Pomc*), cocaine- and amphetamine-regulated transcript (*Cart*)*,* neuropeptide Y (*Npy*)*,* and agouti-related peptide (*Agrp)* mRNA expression levels in the hypothalamus after intracerebroventricular administration of GIP. GIP administration significantly increased *Pomc* mRNA levels (GIP-treated mice: 7.59 ± 0.14; saline-treated mice: 3.44 ± 1.38 arbitrary units; *p* = 0.030). However, after acute intracerebroventricular GIP administration, we did not observe significant differences in *Lepr* (GIP-treated:1.00 ± 0.17; saline-treated: 1.05 ± 0.10 arbitrary units; *p* = 0.827), *Cart* (GIP-treated: 2.35 ± 0.92; saline-treated: 1.44 ± 0.61 arbitrary units; *p* = 0.428), *Npy* (GIP-treated: 0.81 ± 0.06; saline-treated: 1.13 ± 0.29 arbitrary units; *p* = 0.326), and *Agrp* (GIP-treated: 1.21 ± 0.11; saline-treated: 1.05 ± 0.14 arbitrary units; *p* = 0.374) mRNA levels in the hypothalamus.

## 3. Discussion

Our study shows that GIPR expression in ARC and circulation GIP levels in mice differ by feeding status. ARC *Gipr* mRNA expression and plasma GIP levels increased during the fed state in obese mice but not in lean mice. The anti-obesity effect of GIP is likely exerted primarily through ARC in the brain. In addition, acute centrally administered GIP decreased food intake and body weight and increased anorexigenic POMC expression in the hypothalamus of the mouse. These results indicate that GIP regulates appetite and body weight through POMC neurons in the brain.

Increased fasting serum immunoreactive GIP levels in obese individuals and a significant increase in immunoreactive GIP levels owing to a high caloric test meal have been observed in obese individuals [3]. HFD-fed mice exhibited a hypersecretion of GIP. HFD-fed mice lacking GIPR did not develop obesity. Mice with GIPR deficiency gained less body weight and fat mass when challenged with a HFD [5]. Furthermore, the inhibition of endogenous GIP or GIPR confers resistance to diet-induced obesity. Resolution of obesity is seen in GIP-reduced mice under HFD conditions [16]. Ablation of GIP-producing cells in mice reduced HFD-induced obesity [7]. Immunoneutralization of GIP using specific monoclonal antibody attenuates the development of obesity in HFD-fed mice [6]. Vaccination against GIP reduced body weight gain in HFD-fed mice [17]. GIPR antagonist administration suppressed body weight gain in mice with diet-induced obesity [18]. The inhibition of GIPR signaling in adipose tissue decreased body weight and reduced hepatic steatosis in HFD-fed mice [8]. Based on this evidence, GIP is considered an obesity hormone, and GIP deficiency is believed to protect mice from diet-induced obesity.

Recently, chronic administration of the dual GIP and GLP-1 receptor agonist in mice with diet-induced obesity potently decreased body weight and appetite, and these effects were significantly greater compared to that with GLP-1 receptor agonist treatment alone [9]. Likewise, the administration of GIP does not induce obesity although GIP is considered an obesity hormone. A chronic increase in GIP levels in GIP transgenic mice was associated with reduced diet-induced obesity [19]. Activation of hypothalamic GIPR cells reduced food consumption in mice [15]. GIPR agonists administered at an increased frequency and for an extended duration decreased body weight in mice with diet-induced obesity [20]. These data suggest that GIPR agonism has an anti-obesity effect in a pharmacologic context. Therefore, the action of GIP may paradoxically affect body weight because the inhibition of endogenous GIP action or the exogenous administration of supraphysiological doses of GIP has anti-obesity effects. However, the exact mechanism underlying the anti-obesity effect of GIP or the GIPR system remains unclear.

The hypothalamus regulates energy metabolism. The orexigenic NPY/AgRP and the anorexigenic POMC-expressing neurons in the ARC, adjacent to the third ventricle, are first-order neurons in the hypothalamus responding to signals of circulating adiposity and project to the PVH and lateral hypothalamic area (LHA), which are connected with the NTS, a brainstem area that integrates sensory information from the peripheral tissues [21,22]. The GIPR is expressed in various tissues including pancreatic islets and adipose tissue and is distributed throughout the CNS, including in areas regulating energy homeostasis [1,5,13]. *Gipr* mRNA is present in the pancreas, gut, adipose tissue, heart, pituitary, adrenal cortex, cerebral cortex, hippocampus, and olfactory bulb [14]. Widespread distribution of GIP-immunoreactive cells in the adult rat hypothalamus and brainstem has been reported [23]. GIPR cells were identified in the ARC, dorsomedial hypothalamus (DMH), and PVH [15]. A centrally administered antibody neutralizing GIPR reportedly reduced body weight and adiposity in mice with diet-induced obesity [13]. CNS-specific GIPR knockout mice had lower body weights and reduced fat accumulation under HFD feeding. Weight loss effects of intracerebroventricular administration of long-acting (fatty acylated) GIP were attenuated in CNS-specific GIPR knockout mice [12]. These data suggest that the GIPR in the CNS may have a key role in energy metabolism regulation. In our study, ARC *Gipr* mRNA expression increased during the fed state in obese mice; however, PVH and NTS *Gipr* mRNA levels did not differ by feeding status both in lean and obese mice. These data suggest that ARC *Gipr* mRNA expression is positively regulated by food intake in obesity. Increased levels of GIPR expression in the ARC may contribute to increased GIP action in the ARC and may be considered a mechanism to reduce food intake through the GIP in obesity. Therefore, ARC may play a key role for GIP activity in the regulation of food intake and energy balance.

Acute central administration of acyl-GIP improved body weight and food intake in mice with diet-induced obesity and increased cFOS neuronal activity in the ARC, DMH, ventromedial hypothalamus, and LHA [12]. Administration of GIP with lateral cerebroventricular cannulas for 4 days resulted in the upregulation of hypothalamic mRNA levels of *Npy* in rats [24]. Centrally administered GIP diminished hypothalamic sensitivity to leptin and increased hypothalamic levels of the suppressor of cytokine signaling-3, an inhibitor of leptin activity, in mice [13]. In a previous study, intracerebroventricular administration of GIP (6 nmol) decreased food intake and body weight in mice, and intracerebroventricular co-administration of GLP-1 and GIP increased *Pomc* expression in ARC using immunofluorescence staining [11]. Acute centrally administered GIP increased *Pomc* expression in the mouse hypothalamus in our study. These data suggest that POMC may be involved in energy metabolism-related hypothalamic circuits affected by the administration of GIP. POMC neurons are likely the targets of GIP action in the brain.

This is the first study confirming the expression of GIPR in each region of the brain and adipose tissue according to feeding status in lean and obese mice. In particular, only a few reports on the altered expression of genes involved in the regulation of food intake and body energy balance in the hypothalamus after the central administration of GIP have been published. Our results highlight that it is likely that the effects of GIP on energy metabolism occur primarily through the ARC in the brain and that POMC neurons may play an important role in central GIP-GIPR signaling.

Our study had limitations. We only studied *Gipr* mRNA expression and not protein expression. The comparison between the lean (chow diet-fed) group and the obese (HFD-fed) group could be considered inappropriate as we did not feed the control group a specific 10% fat control diet. We could not determine the exact mechanism. We could not ascertain whether the activation of the GIPR or the inactivation of the GIPR can be used for the treatment of obesity. Chronic GIPR agonism desensitizes adipocyte GIPR activity mimicking GIPR antagonism [25]. We did not determine whether the GIPR is expressed in POMC neurons. However, ARC POMC neurons most likely express the GIPR, because GIPR agonists were reported to activate POMC neurons, as ascertained by measuring cytosolic Ca^2+^ concentration [26]. Therefore, the effect of GIP on POMC is likely to be direct. To determine the role of the GIPR in ARC or POMC-expressing neurons, further studies including the specific deletion of the GIPR in hypothalamic POMC or AgRP-expressing neurons are needed.

## 4. Materials and Methods

### 4.1. Animals

Male C57BL/6 mice aged 7 weeks were purchased from Orient Bio (Seongnam-si, Korea). Animals were maintained on a 12:12 h light-dark cycle with lights on from 7 a.m. and free access to water and a standard chow diet (containing 14% of calories from fat; Altromin, Lage, Germany, 1314) or a HFD (60% fat; Research Diet, New Brunswick, NJ, USA, #D12492) for 20 weeks.

### 4.2. Study Design

To determine the *Gipr* mRNA expression in the brain according to feeding status in lean (weighing 27–30 g) and obese (weighing 44–50 g) mice, the mice were divided into 4 groups: fed/lean, fasted/lean, fed/obese, and fasted/obese groups. The mice in the fasted groups underwent a 24 h fast before harvest. In the early light phase (9 a.m.–11 a.m.), the ARC, PVH, NTS in the brainstem, and adipose tissues including gonadal fat, brown adipose tissue, and inguinal fat were collected under isoflurane anesthesia. The harvested tissues were immediately frozen in liquid nitrogen and kept at −80 °C until RNA extraction. To analyze brain region-specific *Gipr* mRNA expression, the ARC, PVH, and NTS were dissected using the micropuncture technique using a stainless-steel coronal 1.0 mm brain matrix (SA-2175, Roboz Surgical Instrument, Gaithersburg, MD, USA) and blunted 18-gauge metal needles (Cellink, Gothenburg, Sweden). The PVH was harvested bilaterally from a 1 mm thick brain slice and the ARC was harvested bilaterally from two consecutive 1 mm thick brain slices [27].

For the acute intracerebroventricular administration test, mice were fasted overnight and administered a single bolus injection of GIP or saline (n = 5–6) at 9 a.m. A syringe pump (Legato 130, KD Scientific, Holliston, MA, USA) was connected to an internal cannula using polyethylene tubing, and the internal cannula was placed into the guide cannula. A total of 2 μL of volume of the GIP or 0.9% saline was delivered through the internal cannula (#C315I/Spc, Protech International, Boerne, TX, USA) using a syringe pump at a rate of 5 μL/min. The mice were handled daily for 1 week before intracerebroventricular injection to acclimatize mice to the intracerebroventricular procedure and they were not restrained or anesthetized during injection. Food intake and body weight changes were monitored for 24 h after GIP (6 nmol) or saline injection. The dosage for intracerebroventricular administration of GIP was selected based on a previous study [11]. After 1 week, the hypothalamic tissue blocks were collected 2 h after intracerebroventricular administration of 6 nmol of GIP or saline. The harvested hypothalamic tissues were immediately frozen in liquid nitrogen and kept at −80 °C until RNA extraction. We dissected the medial part of the hypothalamus in the anterior border of the optic chiasm, the posterior border of the mammillary body, the upper border of the anterior commissure, and the lateral border halfway from the lateral sulcus in the ventral side of the brain. The hypothalamic neuropeptide mRNA levels including those of *Lepr*, *Pomc*, *Cart*, *Npy*, and *Agrp* were determined.

### 4.3. In Vivo Intrahypothalamic Treatment

To investigate the effects of central GIP administration (Tocris, Bristol, UK, #2084) on the relative expression of hypothalamic neuropeptide mRNA levels, we performed stereotaxic surgery on 8-week-old C57BL/6 mice and placed 26-gauge stainless steel cannulas into the third ventricles of the mice. We performed stereotaxic surgery on mice anesthetized using an intraperitoneal injection of 40 mg/kg Zoletil and 5 mg/kg Rompun and placed the stainless steel guide cannulas (26 gauge, #C315G/Spc) into the third ventricles of the mice. The coordinates for implantation were 1.8 mm caudal to the bregma and 5.0 mm ventral to the sagittal sinus using the stereotaxic instrument. Dental cement (Vertex Self Curing) was used to attach the guide cannula to the mouse skull. When GIP was not injected into the mice, dummy cannulas (#C315DC/Spc) were used to occlude the guide cannulas. After 7 days of recovery, 50 ng of angiotensin II (Sigma-Aldrich, Saint Louis, MO, USA, #A9525) was administered at the cannulas to test correct positioning [28]. We excluded mice not exhibiting drinking behavior within 15 min after angiotensin II (50 ng) intracerebroventricular administration.

### 4.4. Gipr, Lepr, Pomc, Cart, Npy, and Agrp mRNA Level Measurements Using Real-Time PCR

Total RNA was extracted using Trizol reagent (Invitrogen, Waltham, MA, USA, #10296010). RNA was treated with DNase/RNase-free distilled water (Invitrogen, #10977015). cDNA was prepared using M-MLV Reverse Transcriptase (Promega, Madison, WI, USA, #M1701). The relative concentrations of *Gipr*, *Lepr*, *Pomc*, *Cart*, *Npy*, and *Agrp* mRNA were corrected using the concentration of *Gapdh* mRNA measured through real-time PCR using primer sets (Appendix A). Relative concentrations of *Gipr* mRNA were calculated from the corrected Ct (cycle of threshold) values (Ct*_Gipr_* − Ct*_Gapdh_*).

### 4.5. Plasma GIP Measurements

Plasma GIP levels were measured using commercially available ELISA kits (Merck, Rahway, NJ, USA, #EZRMGIP-55K).

### 4.6. Statistical Analyses

All data values are presented as mean ± standard error of the mean. Differences between GIP-treated mice and saline-treated mice were analyzed using the unpaired *t* test (Figure 4 and Figure 5) and between fed/lean mice, fasted/lean mice, fed/obese mice, and fasted/obese mice using one-way analysis of variance (ANOVA) with Bonferroni corrections (Figure 1, Figure 2 and Figure 3). A two-way ANOVA was performed to evaluate the interaction effect of two factors (factor 1: obese vs. lean; factor 2: fed vs. fasted). Statistical analysis was performed using the SPSS 23.0 software package for Windows (IBM Corporation, Chicago, IL, USA). *p* < 0.05 was considered statistically significant.

## 5. Conclusions

In conclusion, our results suggest that ARC likely plays an important role in influencing the energy metabolism-regulatory activities of GIP. Specifically, GIP has the potential to regulate POMC-mediated appetite regulation in the hypothalamus. The effects of GIP on energy metabolism appear to occur primarily through the ARC in the brain, and POMC neurons may play an important role in central GIP-GIPR signaling. Additional studies involving the specific deletion of GIPR in hypothalamic POMC-expressing neurons are needed.

## Figures and Tables

**Figure 1 ijms-26-01142-f001:**
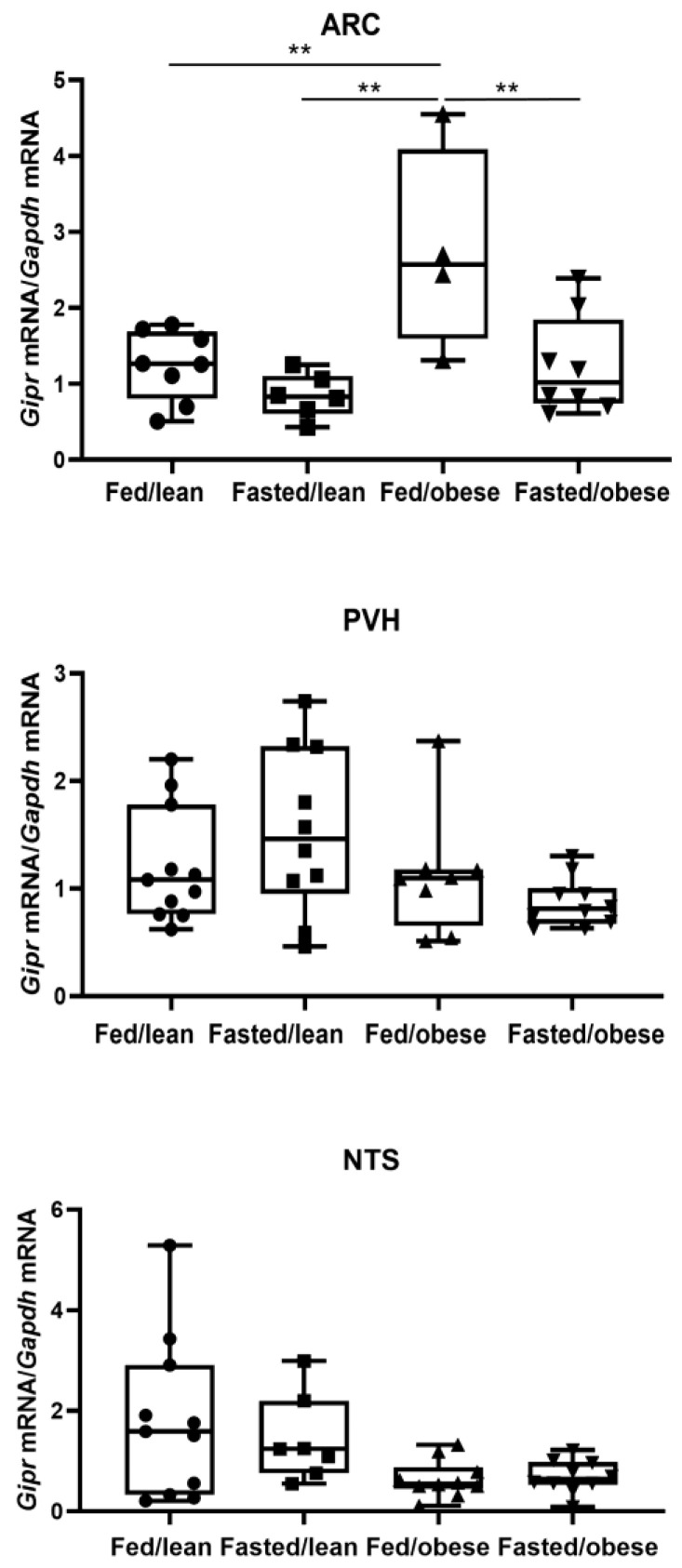
Expression levels of gastric inhibitory polypeptide receptor mRNA in the mouse brain by feeding state. The mice in the fasted groups underwent a 24 h fast before harvest. Gastric inhibitory polypeptide receptor mRNA levels were analyzed using real-time PCR and expressed as gastric inhibitory polypeptide receptor mRNA/*Gapdh* mRNA (n = 4–11 per group). ** *p* < 0.01 between the indicated groups. Differences between the four groups (fed/lean mice, fasted/lean mice, fed/obese mice, and fasted/obese mice) using one-way analysis of variance with Bonferroni corrections. *Gipr*, gastric inhibitory polypeptide receptor; ARC, arcuate nucleus; PVH, paraventricular nucleus of hypothalamus; NTS, nucleus of the solitary tract.

**Figure 2 ijms-26-01142-f002:**
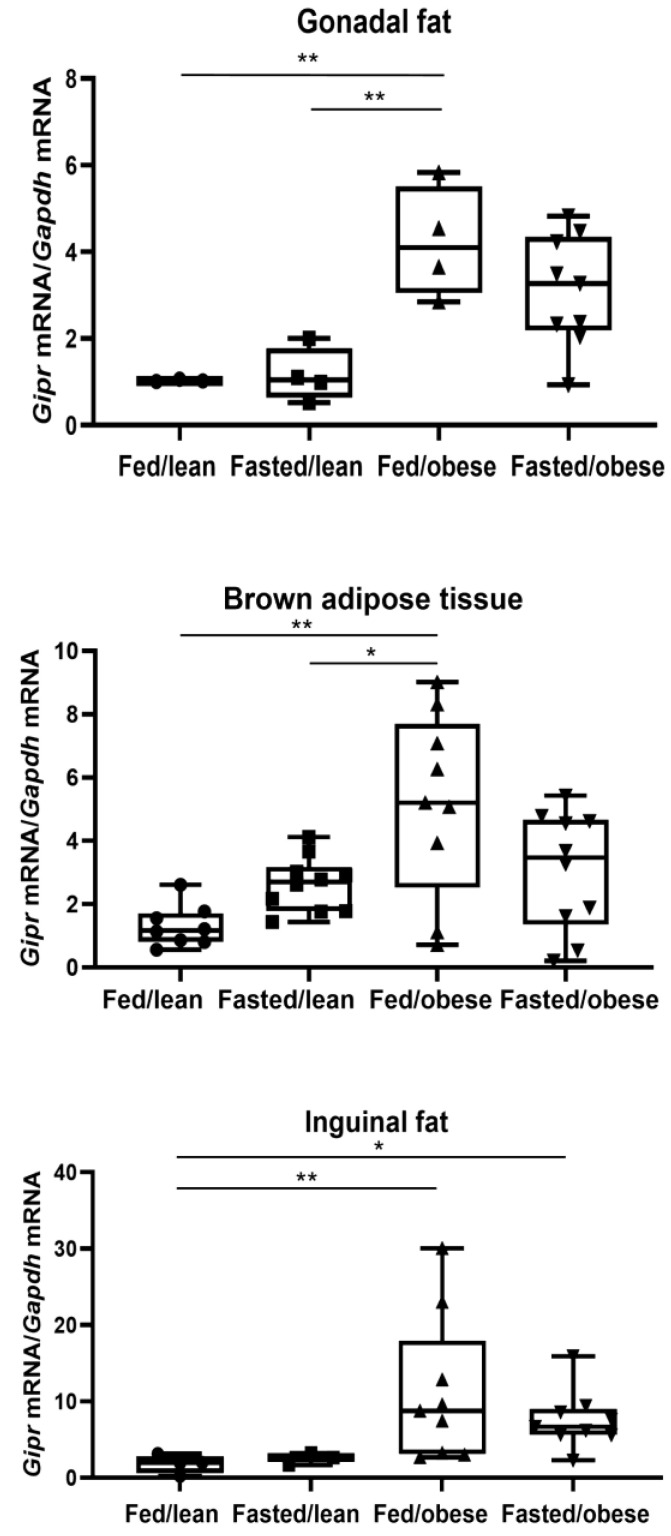
Expression levels of gastric inhibitory polypeptide receptor mRNA in the mouse adipose tissue by feeding status. The mice in the fasted groups underwent a 24 h fast before harvest. Gastric inhibitory polypeptide receptor mRNA levels were analyzed using real-time PCR and expressed as gastric inhibitory polypeptide receptor mRNA/*Gapdh* mRNA (n = 3–9 per group). * *p* < 0.05, ** *p* < 0.01 between the indicated groups. Differences between the four groups (fed/lean mice, fasted/lean mice, fed/obese mice, and fasted/obese mice) using one-way analysis of variance with Bonferroni corrections. *Gipr*, gastric inhibitory polypeptide receptor.

**Figure 3 ijms-26-01142-f003:**
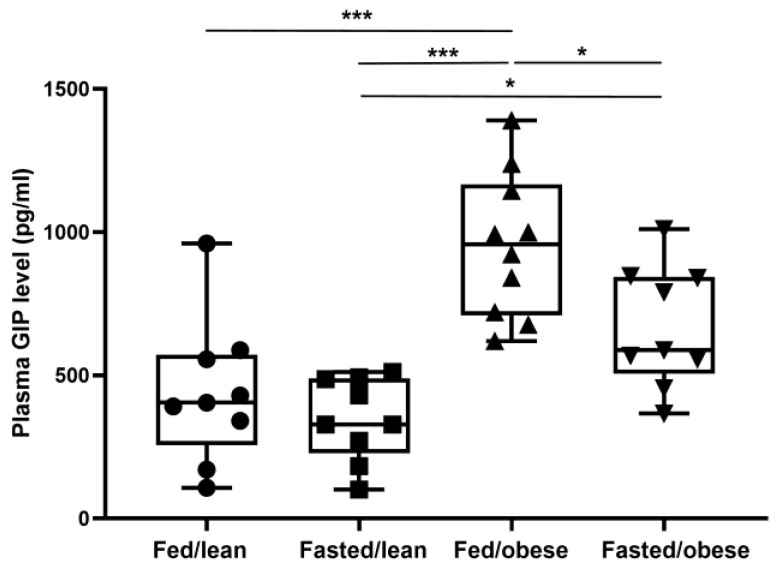
Plasma gastric inhibitory polypeptide levels of the mice by feeding state. * *p* < 0.05, *** *p* < 0.001 between the indicated groups (n = 9–10 per group). Differences between the four groups (fed/lean mice, fasted/lean mice, fed/obese mice, and fasted/obese mice) using one-way analysis of variance with Bonferroni corrections. GIP, gastric inhibitory polypeptide.

**Figure 4 ijms-26-01142-f004:**
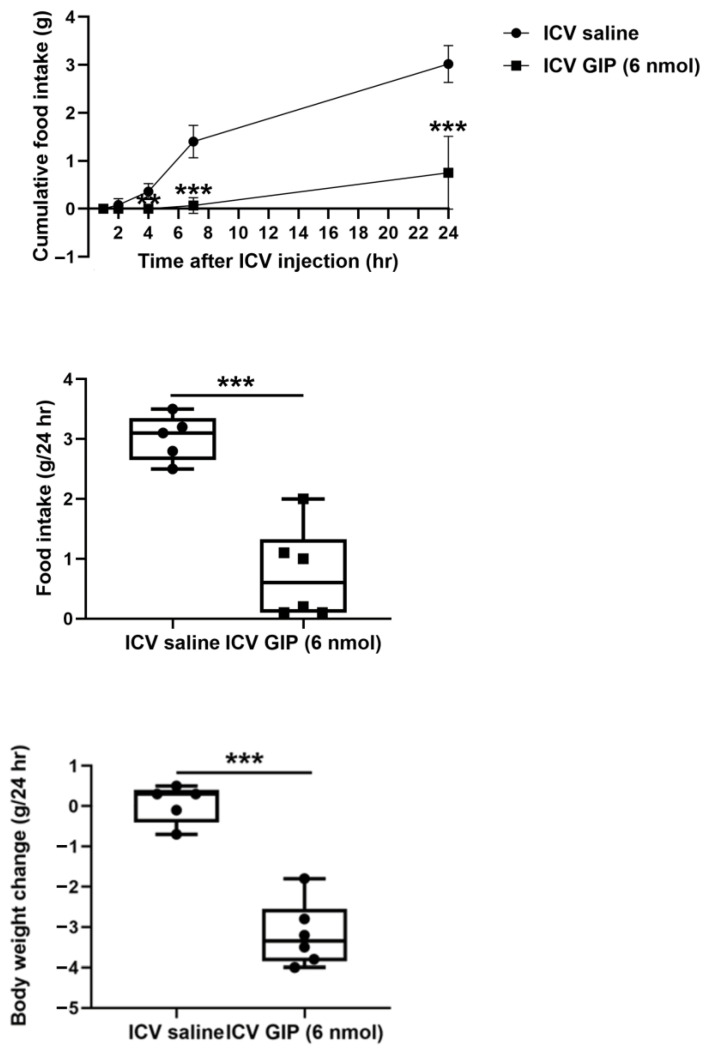
Effect of intracerebroventricular injection of gastric inhibitory polypeptide on cumulative food intake and body weight. ** *p* < 0.01, *** *p* < 0.001 between GIP-treated mice and saline-treated mice (n = 5–6 per group). Differences between GIP-treated mice and saline-treated mice were analyzed using the unpaired t test. ICV, intracerebroventricular; GIP, gastric inhibitory polypeptide.

**Figure 5 ijms-26-01142-f005:**
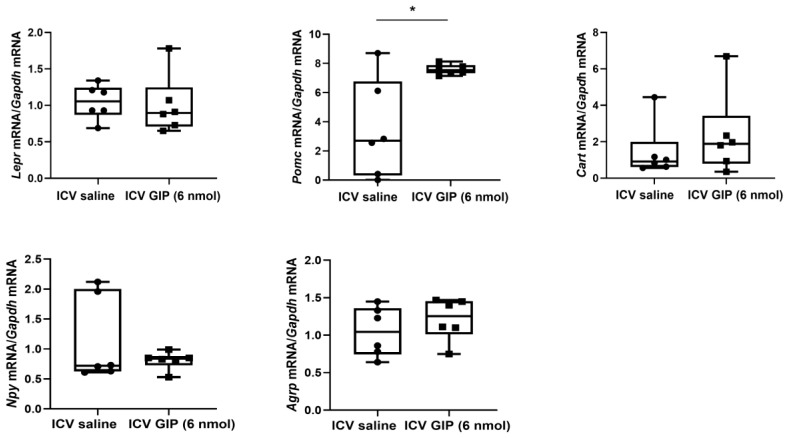
The expression levels of leptin receptor, proopiomelanocortin, cocaine- and amphetamine-regulated transcript, neuropeptide Y, and agouti-related peptide mRNA in the hypothalamus after intracerebroventricular injection of gastric inhibitory polypeptide. * *p* < 0.05 between GIP-treated mice and saline-treated mice (n = 5–6 per group). Differences between GIP-treated mice and saline-treated mice were analyzed using the unpaired *t* test. ICV, intracerebroventricular; GIP, gastric inhibitory polypeptide.

## Data Availability

All data generated or analyzed during this study are included in this article.

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
