# Peer review of "Differences in GIP Receptor Expression by Feeding Status in the Mouse Brain"

_ijms, 2025, doi:10.3390/ijms26031142_

Round 1
Reviewer 1 Report
Comments and Suggestions for Authors
After a comprehensive
evaluation of the paper, I decided to reject the manuscript and not encourage re-submission. We understand have hope but it is very poor quality. I hope that authors will improve paper and will submit it elsewhere. It can not be process this manuscript in this redaction.
Comments on the Quality of English Language
After a comprehensive
evaluation of the paper, I decided to reject the manuscript and not encourage re-submission. We understand have hope but it is very poor quality. I hope that authors will improve paper and will submit it elsewhere. It can not be process this manuscript in this redaction.
Author Response
Thank you for your time and consideration in reviewing this manuscript.
This is the first study confirming the expression of GIPR in each region of the brain and adipose tissue according to feeding status in lean and obese mice. In particular, only few reports on the altered expression of genes involved in regulation of food intake and body energy balance in the hypothalamus after central administration of GIP have been published. Our results highlight that it is likely that the effects of GIP on energy metabolism likely occur primarily through the ARC in the brain and that POMC neurons may play an important role in central GIP-GIPR signaling.
Reviewer 2 Report
Comments and Suggestions for Authors
241226_Comments
The manuscript submitted by Dr. Song et al. described the mechanism of GIPR regulating the energy balance in the CNS. The authors found that ARC GIPR is affected by feeding conditions, and the target gene of POMC was identified. This study is scientifically sound, and the manuscript's performance is good. Some minor concerns are needed to be improved:
1. The preferred name of GIP is gastric inhibitory polypeptide. The authors are recommended to revise it in the manuscript.
2. Line 254: “The mice were handled daily for 1 week prior to ICV injection to acclimatize mice to the ICV procedure and they were not restrained or anaesthetized during injection.”
=> Was the treatment performed with only saline for one week?
3. Does Figure 5 show the results for 4.3. In Vivo Intrahypothalamic Treatment? Did the authors use lean mice?
4. Gene symbols for mice should be italicized, with the first letter in upper case and all the rest in lower case. For example, Gipr.
For more details, https://www.informatics.jax.org/mgihome/nomen/short_gene.shtml
Author Response
- The preferred name of GIP is gastric inhibitory polypeptide. The authors are recommended to revise it in the manuscript.
→ We revised the manuscript as you recommended.
- Line 254: “The mice were handled daily for 1 week prior to ICV injection to acclimatize mice to the ICV procedure and they were not restrained or anaesthetized during injection.”
=> Was the treatment performed with only saline for one week?
→ We removed and reinserted the dummy cannula at the same time each day without infusing any substance into the mice through the guide cannula.
- Does Figure 5 show the results for 4.3. In Vivo Intrahypothalamic Treatment? Did the authors use lean mice?
→ Yes, we used lean mice for in vivo hypothalamic treatment.
- Gene symbols for mice should be italicized, with the first letter in upper case and all the rest in lower case. For example, Gipr.
For more details, https://www.informatics.jax.org/mgihome/nomen/short_gene.shtml
→Thank you for your advice. We modified the gene symbols of the mice.
Reviewer 3 Report
Comments and Suggestions for Authors
The reviewed manuscript examines the levels of glucose-dependent insulinotropic polypeptide (GIP) in the brain of fast and fed mice. The paper is well written with clear objectives. Yet, improvements are recommended as follows:
In the abstract, mention the dose of GIP used for ICV injection. Define POMC in terms of neuronal function.
In the introduction, refer to the expression of GIP in other tissues, including adipose tissue. Comment briefly of cellular localization of GIP-GIPR and what is already known about them in terms of molecular interaction.
In all figures, clear what you mean by “the two groups” and “4 groups." Name them! Also were the error bars referring to SD or SE?
Figure 1. Define abbreviations on the figure within the legend.
Figure 2. Write fasting duration in the figure legend.
Figure 3. Write ICV in full
Figure 4, 5. Did you use any post hoc tests for statistical analysis?
Figure 5. Write gene names in full.
In the study design (4.2.), add references for sampling different brain parts.
The primer sequence (4.4) better be shown in table form.
The conclusion is very brief. Please extend by elaboration of the significance of GIP-GIPR signaling. Add future directions of this work.
Author Response
The reviewed manuscript examines the levels of glucose-dependent insulinotropic polypeptide (GIP) in the brain of fast and fed mice. The paper is well written with clear objectives. Yet, improvements are recommended as follows:
In the abstract, mention the dose of GIP used for ICV injection. Define POMC in terms of neuronal function.
→ As you advised, we mentioned the dose of GIP used for ICV injection.
→ “It is possible that POMC neurons are targets of GIP action in the brain.” was inserted in the abstract.
In the introduction, refer to the expression of GIP in other tissues, including adipose tissue. Comment briefly of cellular localization of GIP-GIPR and what is already known about them in terms of molecular interaction.
→ “GIP receptor mRNA was reported to be present in peripheral organs and the brain including pancreas, gut, adipose tissue, heart, pituitary, and the adrenal cortex.” was inserted in the introduction.
“GIPR signaling in adipose tissue has been suggested to play an important role in HFD-induced insulin resistance and hepatic steatosis through adipose tissue-specific GIPR knockout mice.” was inserted in the introduction.
In all figures, clear what you mean by “the two groups” and “4 groups." Name them! Also were the error bars referring to SD or SE?
→ In figures 1-3, the two groups represent fed lean mice and fasted lean mice, fed lean mice and fed obese mice, fed lean mice and fasted obese mice, fasted lean mice and fed obese mice, fasted lean mice and fasted obese mice, and fed obese mice and fasted obese mice. In figures 1-3, the four groups represent fed lean mice, fasted lean mice, fed obese mice, and fasted obese mice.
In figures 4-5, the two groups represent GIP-treated mice and saline-treated mice.
Error bars represent standard error values.
Figure 1. Define abbreviations on the figure within the legend.
→ As you recommended, we included the abbreviations in the legend.
Figure 2. Write fasting duration in the figure legend.
→ As you recommended, we include fasting duration in the figure legend. “The mice in the fasted groups underwent a 24 h fast before harvest.” was inserted in the figure legend.
Figure 3. Write ICV in full
→ As you suggested, we changed “ICV” to “intracerebroventricular”.
Figure 4, 5. Did you use any post hoc tests for statistical analysis?
→ Differences between two groups were analyzed using the unpaired t test (Fig 4, 5) without any post hoc tests for statistical analysis.
Figure 5. Write gene names in full.
→ We revised the manuscript as you recommended.
In the study design (4.2.), add references for sampling different brain parts.
→ As recommended, we added references for sampling different brain parts.
The primer sequence (4.4) better be shown in table form.
→We tabulated primer sequences separately as recommended.
Supplementary Table 1. Primer sequences used for real time PCR analysis
|
Gene |
Forward |
Reverse |
|
Gipr |
5’- CTGCCTGCCGCACGGCCCAGAT-3’ |
5’- GCGAGCCAGCCTCAGCCGGTAA-3’ |
|
Lepr |
5’- GATTTCTTGGGACAGCCAAA-3’ |
5’- TCCAGACTCCTGAACCATCC-3’ |
|
Pomc |
5’- CTAAGAGAGGCCACTGAACA-3’ |
5’- TCTATGGAGGTCTGAAGCAG-3’ |
|
Cart |
5’- TGCTGGGATTAAAGGCGTGT-3’ |
5’-TCTCTGAGGGGAACGCAAAC-3’ |
|
Npy |
5’- GGTGGATCTCTTCTCTCACA-3’ |
5’- CAGAGCGGAGTAGTATCTGG-3’ |
|
Agrp |
5’-TAGGTGCGACTACAGAGGTT-3’ |
5’- GAGGTGCTAGATCCACAGAA-3’ |
The conclusion is very brief. Please extend by elaboration of the significance of GIP-GIPR signaling. Add future directions of this work.
→ Thanks for your comment. ”The effects of GIP on energy metabolism appear to occur primarily through the ARC in the brain, and POMC neurons may play an important role in central GIP-GIPR signaling. Additional studies involving specific deletion of GIPR in hypothalamic POMC-expressing neurons are needed. “ were inserted in the conclusion.